# The V2 Protein from the Geminivirus Tomato Yellow Leaf Curl Virus Largely Associates to the Endoplasmic Reticulum and Promotes the Accumulation of the Viral C4 Protein in a Silencing Suppression-Independent Manner

**DOI:** 10.3390/v14122804

**Published:** 2022-12-15

**Authors:** Liping Wang, Pengfei Fan, Tamara Jimenez-Gongora, Dan Zhang, Xue Ding, Laura Medina-Puche, Rosa Lozano-Durán

**Affiliations:** 1Shanghai Center for Plant Stress Biology, CAS Center for Excellence in Molecular Plant Sciences, Chinese Academy of Sciences, Shanghai 201602, China; 2Department of Plant Biochemistry, Centre for Plant Molecular Biology (ZMBP), Eberhard Karls University, D-72076 Tübingen, Germany

**Keywords:** virus, geminivirus, endoplasmic reticulum (ER), tomato yellow leaf curl virus (TYLCV), V2, ERQC, protein accumulation, C4, myristoylation, NMT1

## Abstract

Viruses are strict intracellular parasites that rely on the proteins encoded in their genomes for the effective manipulation of the infected cell that ultimately enables a successful infection. Viral proteins have to be produced during the cell invasion and takeover in sufficient amounts and in a timely manner. Silencing suppressor proteins evolved by plant viruses can boost the production of viral proteins; although, additional mechanisms for the regulation of viral protein production likely exist. The strongest silencing suppressor encoded by the geminivirus tomato yellow leaf curl virus (TYLCV) is V2: V2 suppresses both post-transcriptional and transcriptional gene silencing (PTGS and TGS), activities that are associated with its localization in punctate cytoplasmic structures and in the nucleus, respectively. However, V2 has been previously described to largely localize in the endoplasmic reticulum (ER), although the biological relevance of this distribution remains mysterious. Here, we confirm the association of V2 to the ER in *Nicotiana benthamiana* and assess the silencing suppression activity-independent impact of V2 on protein accumulation. Our results indicate that V2 has no obvious influence on the localization of ER-synthesized receptor-like kinases (RLKs) or ER quality control (ERQC)/ER-associated degradation (ERAD), but dramatically enhances the accumulation of the viral C4 protein, which is co-translationally myristoylated, possibly in proximity to the ER. By using the previously described V2_C84S/86S_ mutant, in which the silencing suppression activity is abolished, we uncouple RNA silencing from the observed effect. Therefore, this work uncovers a novel function of V2, independent of its capacity to suppress silencing, in the promotion of the accumulation of another crucial viral protein.

## 1. Introduction

Viruses are obligate intracellular parasites that strictly depend on the molecular machinery of the infected cells in order to multiply and spread. During the infection cycle, different viruses can co-opt or interfere with multiple cellular processes occurring in diverse subcellular localizations. In animals, the endoplasmic reticulum (ER) plays a central role in virus–host interactions, and viruses from different families converge in that they have developed strategies to manipulate this organelle to promote infection (reviewed in [1,2]). In contrast, in plants, the understanding of the role the ER plays in viral infections is limited, although this organelle is also emerging as a target for viruses in this kingdom (reviewed in [3,4]).

The ER is an organelle made of an endomembrane system, comprising the outer nuclear envelope and a connected intricate network of tubules and sheets referred to as the peripheral ER. A major function of the ER is the synthesis of membrane-localized and secreted proteins. Once these proteins are synthesized, ER-resident enzymes assist in protein folding and apply organelle-specific post-translational modifications, such as glycosylation. Properly formed proteins exit the ER in COPII-coated vesicles; misfolded proteins, however, will accumulate in the ER, be perceived by the ER quality control (ERQC) surveillance system, and trigger ER stress, which is relayed to the nucleus via the Unfolded Protein Response (UPR), resulting in an increase in the availability of ER chaperones and a momentary stop of translation. If the UPR is not successful in triggering the proper folding of the client protein, this will be translocated to the cytosol to be degraded by the proteasome in a pathway called ER-associated degradation (ERAD) [5]. 

Proteins encoded by plant viruses have been shown to activate the expression of ER stress-related genes. That is the case of triple gene block protein 3 (TGBp3) from potato virus X (PVX), triple gene block 3 (TGB3) from Plantago asiatica mosaic virus (PlAMV), 6K2 from turnip mosaic virus (TuMV), P10 from rice black-streak dwarf virus, and p11 from garlic virus X [6,7,8,9,10,11]. Infection by soybean mosaic virus (SMV) activates the UPR, and this promotes viral accumulation [12]; interestingly, the P3 protein encoded by SMV localized in the ER acts as a virulence factor and might activate the UPR through its interaction with the plant eEF1A [12,13]. TGBp3 resides in the ER and promotes the expression of ER-resident chaperones in both *Arabidopsis thaliana* and *Nicotiana benthamiana* ([10,11,14]), a phenomenon that has been hypothesized to enable massive viral protein synthesis [15,16,17]; to date, however, the exact molecular mechanism underlying the activation of the UPR by TGBp3 remains to be determined.

The V2 protein encoded by the DNA geminivirus tomato yellow leaf curl virus (TYLCV) has been described as largely associated to the ER [18]; the functions of V2 described so far, however, seem to rely on other subcellular localizations. V2 from TYLCV is a strong suppressor of post-transcriptional gene silencing (PTGS) through its interaction with SGS3 [19,20], which occurs in punctate structures in the cytoplasm [20] and suppresses transcriptional gene silencing (TGS) through its interactions with HDA6 and AGO4 in the nucleus and the Cajal body, respectively [21,22]. V2 has also been recently described to mediate the nuclear export of the capsid protein (CP) [23]. The biological role(s) of the ER association of V2 to date, therefore, remains elusive.

In this work, we confirm the ER-association of V2 from TYLCV using confocal microscopy and test its potential impact on the localization and accumulation of plasma membrane-localized receptor-like kinases (RLKs) and on ERQC/ERAD. Strikingly, we found that V2 can promote the accumulation of other virus-encoded proteins, in particular C4, and that this effect can be uncoupled from its PTGS suppression activity. Our results, therefore, uncover a novel biological role of V2 during the viral infection, which might be connected with its subcellular localization at the ER.

## 2. Materials and Methods

### 2.1. Plasmids and Cloning 

Constructs to express GFP, V2-GFP, GFP-V2, Rep-GFP, C4-GFP, C4_G2A_-GFP, p19, BAM1-GFP, FLS2-GFP, or NIK1-GFP have been previously described [24,25,26,27]. 

Using the pENTR-D/TOPO-V2 (with stop codon) as a template [24], the V2_C84S/C86S_ mutant, carrying two C-to-S mutations in the 84th amino acid (converting TGC to AGC) and 86th amino acid (converting TGT to AGT) of the V2 protein, was generated with the Quick Change Lightning Site-Directed Mutagenesis Kit (Cat #210518, Agilent Technologies, Santa Clara, CA, USA) following the manufacturer’s instructions. Then, V2_C84S/C86S_ was subsequently Gateway-cloned to binary constructs pGWB2, pGWB505, and pGWB506 [28,29] to express V2_C84S/C86S,_ V2_C84S/C86S_-GFP, and GFP-V2_C84S/C86S_, respectively.

pENTR-D/TOPO-GFP was generated in previous work [24]. The binary construct to express RFP-GFP was generated by Gateway-cloning (LR reaction, Cat# 11791100, Invitrogen™, Thermo Scientific) the GFP from pENTR/D-TOPO into pGWB555 [28]. Constructs expressing BRI-GFP, bri1-9-GFP, and bri1-5-GFP are described in [30], and the construct to express RFP-HDEL is described in [31]. The *NMT1* CDS from Arabidopsis (AT5G57020) was obtained by reverse transcription-polymerase chain reaction (RT-PCR) and cloned in pENTR-D/TOPO (Cat# K240020SP, Invitrogen™, Thermo Scientific, Waltham, MA, USA) with a stop codon. Then NMT1 were subsequently Gateway-cloned into the binary construct pGWB506 [28] to express GFP-NMT1.

All primers and plasmids used for cloning are summarized in Appendix A, respectively.

### 2.2. Plant Materials and Growth Conditions

*N. benthamiana* plants were grown in a controlled growth chamber under long-day conditions (LD, 16 h light/8 h dark) at 22 °C or 25 °C, respectively.

### 2.3. Bacterial Strains and Growth Conditions

*Escherichia coli* strain DH5α was used for general cloning and subcloning procedures, and strain DB3.1 was used to amplify Gateway-compatible empty vectors.

For in planta expression, *Agrobacterium tumefaciens* strain GV3101 harbouring the corresponding binary vectors were liquid-cultured in LB medium (1% tryptone, 0.5% yeast extract, and 1% NaCl) with the appropriate antibiotics at 28°C overnight. 

### 2.4. A. tumefaciens-Mediated Transient Gene Expression in N. benthamiana

Transient expression assays were performed as previously described [21] with minor modifications. In brief, all binary plasmids were transformed into *A. tumefaciens* strain GV3101. *A. tumefaciens* clones carrying the constructs of interest were liquid-cultured in LB with the appropriate antibiotics at 28 °C overnight. Bacterial cultures were then centrifuged at 4000× *g* for 10 min and resuspended in the infiltration buffer (10 mM MgCl_2_, 10 mM MES pH 5.6, 150 μM acetosyringone) and adjusted to an OD_600_ = 0.25, with the exception of the culture to express V2_C84SC86S_ in Figure 3C,D, which was adjusted to 0.75 in order to counter the previously observed lower accumulation of the protein. Next, the bacterial suspensions were incubated in the infiltration buffer at room temperature and in the dark for 2–4 h and then infiltrated into four-week-old *N. benthamiana* plants. For co-expression experiments, the different *A. tumefaciens* suspensions were mixed at a 1:1 ratio before infiltration. 

### 2.5. Visualization of Protein Subcellular Localization

To determine protein subcellular localization, plant tissues expressing GFP-fused proteins were imaged with a Leica TCS SP8 point scanning confocal microscope (Leica Microsystems) using the preset settings for GFP (Ex: 488 nm, Em: 500–550 nm). Confocal imaging for co-localization of GFP, V2-GFP, V2_C84SC86S_-GFP, or GFP-NMT1 and RFP-HDEL were performed using the pre-set sequential scan settings for GFP (Ex: 488 nm, Em: 500–550 nm) and RFP (Ex: 561 nm, Em: 580–630 nm).

### 2.6. Plasmolysis Assay

GFP, V2-GFP or GFP-V2, and RFP-HDEL, or GFP-fused V2_C84SC86S_ alone were transiently expressed in four-week-old *N. benthamiana* leaves by *A. tumefaciens*-mediated infiltration, and 1 M NaCl was infiltrated into *N. benthamiana* leaves expressing the proteins of interest. Plasmolysis was observed under the confocal microscope 15 min after NaCl treatment.

### 2.7. Protein Extraction and Western Blot Assays

Protein extraction was performed as previously described [24] with minor modifications. In brief, proteins were transiently expressed in four-week-old *N. benthamiana* leaves by *A. tumefaciens* -mediated infiltration. Samples were taken two days post infiltration (dpi). After grinding the plant tissue in liquid nitrogen, total proteins were extracted by adding lysis buffer (100 mM Tris-HCl pH 8.0; 150 mM NaCl; 10% glycerol; 5 mM EDTA; 5 mM DTT, 1 mM PMSF; 1% protease inhibitor cocktail; 1% NP-40) and incubating at 4 °C for 15 min. Then the supernatant was carefully collected after the samples were centrifuged at 15,000× *g* at 4 °C for 15 min.

Proteins were separated by SDS/PAGE 10% and analyzed by western blot as previously described [32]. For western blot, the following primary and secondary antibodies were used: mouse anti-green fluorescent protein (GFP) (M0802-3a, Abiocode, Agoura Hills, CA, USA) (1:5000), custom-made rabbit anti-V2 (ABclonal with V2_1-116aa_) (1:2000), goat anti-mouse coupled to horseradish peroxidase (A2554, Sigma, St. Louis, MO, USA) (1:15,000), and goat anti-rabbit coupled to horseradish peroxidase (A0545, Sigma, St. Louis, MO, USA) (1:15,000).

### 2.8. Reverse Transcription Quantitative PCR (RT-qPCR)

To detect Rep, C4, C4_G2A_, and GFP transcripts, total RNA was extracted from *N. benthamiana* leaves by using the Plant RNA kit (OMEGA Bio-tek # R6827). RNA was reverse-transcribed into cDNA by using the iScriptTM cDNA Synthesis Kit (Bio-Rad #1708890) according to the manufacturer’s instructions. *Elongation factor-1 alpha* (*NbEF1a*) was used as a reference gene [33]. Relative expression was calculated by the comparative Ct method (2^−ΔΔCt^). RT-qPCR was performed in a BioRad CFX96 real-time system as described previously [34]. All primers used for RT-qPCR are listed in Appendix A.

### 2.9. Chemical Induction of Cell Death (with BFA or Tunicamycin) and Ion Leakage Measurement Assay

Four-week-old *N*. *benthamiana* plants were infiltrated with *A*. *tumefaciens* carrying the constructs of interest. One day after *A*. *tumefaciens* infiltration, the same leaves were infiltrated with 30 μg/mL Brefeldin A (BFA; B7651, Sigma; ER stress/cell death inducer), 20 μg/mL tunicamycin (TM; ab120296, Abcam; ER stress/cell death inducer), or 0.2% DMSO (mock treatment).

The extent of cell death was measured quantitatively by monitoring electrolyte leakage. Four leaf discs (diameter, 1 cm) of infiltrated leaves were punched out for each sample immediately after BFA/TM/DMSO infiltration (at 1 dpi of *A. tumefaciens* infiltration). Then, the leaf discs were placed in a 12-well plate containing 5 mL of distilled water per well for 1 h to remove ions released by sampling-related injury. After washing, the leaf discs were carefully transferred to a fresh 12-well plate containing 5 mL of distilled water per well, which was incubated in a growth chamber for 24 h. Conductivity measurements were taken with ORION STAR A212 conductivity meter (Thermo Scientific) 24 h after treatment.

## 3. Results

### 3.1. The V2 Protein from TYLCV Localizes to the ER

In TYLCV, the V2 protein has been described as a strong viral silencing suppressor [19,20,21,22] and as potentially involved in viral movement [35], although this is controversial [36]. A recent study has also attributed to V2 a role in the nuclear export of the CP [23]. Intriguingly, V2 has been described as localized in the endoplasmic reticulum in tobacco protoplasts [18]; however, this localization does not seem related to the protein’s previously described functions, and, therefore, its biological relevance remains unknown. In order to determine whether this localization can also be observed upon transient expression in *N. benthamiana*, our usual model plant in the study of plant–TYCLV interactions, we observed transiently transformed *N. benthamiana* leaf patches co-expressing GFP-V2 or V2-GFP [24] and the ER marker RFP-HDEL [31]. As shown in Figure 1A–D, confocal microscopy and co-localization analyses confirm that V2 fused to GFP at the N- or the C-terminus accumulates partially associated to the ER in *N. benthamiana*. In maximum intensity projection views from z-stacks images, it becomes apparent that V2 accumulates in the cell in a pattern different to that of free GFP, including around the nuclei (Figure 1C, Appendix A). Plasmolysis assays confirm the localization of V2-GFP and GFP-V2 inside the cell (Figure 1D). Plants expressing free GFP or RFP-fused GFP were used as a negative control or positive control in co-localization experiments, respectively (Figure 1A–D).

### 3.2. V2 Does Not Affect the Distribution of ER-Retained Proteins nor Cell Death upon Chemically-Induced ER Stress

The spatial association of V2 with the ER prompted us to investigate if this virulence factor may affect the accumulation or secretion of plasma membrane-resident proteins, which are synthesized by the ribosomes in contact with the ER and enter the secretory pathway. For this purpose, we selected a number of RLKs, transmembrane proteins localized at the plasma membrane, to test their accumulation and subcellular localization, since RLKs play important roles in plant development and interactions with the environment, and several of them have been shown to be bound by geminivirus-encoded proteins, including the TYLCV-encoded C4 protein [25,26,37]. The immune flagellin receptor FLAGELLIN SENSING 2 (FLS2), the brassinosteroid receptor BRASSINOSTEROID INSENSITIVE 1 (BRI1), and the geminivirus-targeted NIK1 and BARELY ANY MERISTEM 1 (BAM1) were fused to GFP at their C-terminus and transiently expressed in *N. benthamiana* in the presence or absence of V2. With the aim to evaluate a possible effect of PTGS suppression, results obtained with V2 were compared to those obtained with a V2_C84S/86S_ mutant, deficient in PTGS suppressing activity [20], or with the well-known PTGS suppressor from tombusviruses p19 [38]. Importantly, V2_C84S/86S_ presents a subcellular localization indistinguishable from that of the wild-type protein (Appendix A). As shown in Figure 2, the co-expression of the RLKs with an efficient silencing suppressor (V2, p19) results in their increased accumulation, measured as GFP fluorescence, in varying degrees, an effect not observed upon co-expression with the silencing suppression-deficient V2_C84S/86S_. The presence of the viral proteins, however, does not noticeably affect the subcellular distribution of the RLKs.

In order to assess the potential impact of V2 on ERQC, and in particular on ERAD, we took advantage of the *BRI1* alleles *bri1-5* and *bri1-9*, two ER-retained mutant variants [30,31,39,40]. Co-expression of bri1-5 or bri1-9 fused to GFP at their C-terminus with V2, V2_C84S/86S_, or p19 did not result in obvious changes in their accumulation in the ER (Figure 2B), suggesting that these proteins do not affect ERQC/ERAD, at least in the case of bri1-5 and bri1-9.

Different chemical inducers of ER stress have been described to date, including BFA and tunicamycin. BFA blocks protein transport from ER to Golgi, while tunicamycin inhibits N-linked protein glycosylation, both instances causing ER stress. Treatment with either chemical ultimately results in cell death [41,42] which can be quantitatively measured as ion leakage [43]. To evaluate the potential effect of V2 on ER stress, we determined ion leakage of *N. benthamiana* leaf discs transiently expressing V2, V2_C84S/86S_, or transformed with empty vector as control. As shown in Appendix A, the expression of either version of the viral protein did not have any impact on the ion leakage resulting from treatment with these ER stress inducers.

### 3.3. V2 Promotes the Accumulation of Another Viral Protein Independently of Its Function as a Silencing Suppressor 

Despite its association to the ER [18] (Figure 1), V2 was not found to affect the accumulation or secretion of plasma membrane proteins. It has been proven, however, that ER-associated ribosomes can also translate soluble, cytosolic proteins [44,45,46], hence expanding the potential impact of ER-localized V2 on protein synthesis. Interestingly, the C4 protein from TYLCV is myristoylated [25,47], and NMT1, the myristoyl transferase catalyzing this modification [47], localizes to the ER [48] (Appendix A); since N-terminal myristoylation is a co-translational modification, this suggests that C4 is synthesized in close proximity to the ER. Therefore, we sought to test whether V2 may affect the accumulation of co-expressed C4. For comparison, we included free GFP, the TYLCV-encoded Rep protein, and the C4_G2A_ mutant, in which the N-myristoylation site is mutated, rendering the protein non-myristoylable [25]. Myristoylated C4 associates to the plasma membrane, while interference with this modification leads to an exclusive chloroplast localization of the viral protein [25,47] (Figure 3A). While the WT V2 increased the accumulation of all co-expressed proteins, most likely due to its activity as a silencing suppressor, V2_C84S/86S_ only had a dramatic effect on the accumulation of C4 (Figure 3A,C,D). Importantly, the relative amount of the transcripts encoding Rep, C4, C4_G2A_, or GFP was increased by V2, in agreement with the PTGS suppression mediated by this protein, but not by V2_C84S/86S_, once again ruling out such activity for this mutant version (Figure 3B). Therefore, V2 can induce accumulation of the viral C4 protein independently of its activity as a silencing suppressor, and likely through a novel function linked to its ER association.

## 4. Discussion

Viruses rely on the proteins encoded in their genomes for the effective manipulation of the infected cell: these proteins have to be produced in sufficient amounts during the cell invasion and takeover in a timely manner. Viral silencing suppressors can boost the production of viral proteins, hence acting as virulence factors. Here, however, we describe a novel role of V2, the strongest silencing suppressor from TYLCV, which is independent of its previously described function in countering PTGS, in the promotion of the accumulation of another viral protein, namely C4. C4 is essential for viral infection and can suppress plant anti-viral defence at different levels [25,47]. It is noteworthy that V2 and C4 have also been shown to physically interact, hinting at the potential concerted functions of these two viral proteins during infection [49].

In this work, we explore the localization of V2 associated to the ER and its potential impact on the biology of the infected cell. For this purpose, we tested the co-localization of GFP-V2 or V2-GFP with the ER marker RFP-HDEL and confirmed an association of the viral protein to this organelle. Then, we tested the effect of co-expressing V2 on a number of selected RLK, including two BRI1 mutants which activate ERQC and are degraded by ERAD. Our results indicate that, besides the silencing suppression effect, which visibly increases the accumulation of the tested RLKs in some cases, no obvious effect of V2 can be observed. The situation is drastically different, however, when the protein co-expressed with V2 is the virus-encoded C4. C4 is co-translationally myristoylated, a process that possibly occurs in close physical proximity to the ER, given the subcellular localization of the myristoyl transferase catalyzing this reaction, NMT1. These observations suggest that at least a proportion of C4 may be translated near the ER. Myristoylated C4 localizes at the plasma membrane and in plasmodesmata, where it interacts with the RLK BAM1, interfering with the cell-to-cell spread of RNA silencing [25]. Upon perception of a biotic threat at the cell surface, or during the viral infection, C4 re-localizes to chloroplasts, from where it suppresses the activation of downstream salicylic acid-mediated defences [47]. Strikingly, the presence of V2 leads to a noticeable increase in the accumulation of C4, an effect that is independent of the silencing suppression activity of the former, since V2_C84S/86S_, a mutant version impaired in PTGS suppression, exerts an effect indistinguishable from that of the wild-type V2. Clear plasma membrane localization (Figure 3A) demonstrates that the C4 population produced upon co-expression with V2 is indeed myristoylated since in the absence of this lipidation the viral protein localizes to chloroplasts [25,47]. The molecular mechanism underlying this effect of V2 remains, however, elusive, and, at this point, whether V2 is affecting C4 synthesis or stability is also unknown. Interestingly, V2 has been shown to co-immunoprecipitate proteins involved in protein folding when expressed in *N. benthamiana* [24].

This observed impact on the accumulation of C4 raises the possibility that V2 might promote the translation and/or stability of additional proteins synthesized in proximity to the ER, including myristoylated plant proteins. This hypothetical effect of V2 might influence the outcome of the virus-plant interaction, since myristoylation has been proposed to lay at the crossroads between plants and pathogens [50]. Interestingly, this potential broader effect of V2 would also open the possibility to use this viral protein in biotechnological applications, with the aim of boosting the production of myristoylated proteins of interest in planta.

In summary, this work uncovers a novel function of V2 during the viral infection, potentially linked to its ER-associated localization, facilitating an increase in the accumulation of the virus-encoded, plasma membrane-localized myristoylated C4 protein. Considering the relevance of C4 during the infection, this additional effect of V2 on the plant cell might have a large impact on virulence. Further work will be required to elucidate the molecular underpinnings and cellular determinants of this latest addition to the functional portfolio of the versatile V2.

## Figures and Tables

**Figure 1 viruses-14-02804-f001:**
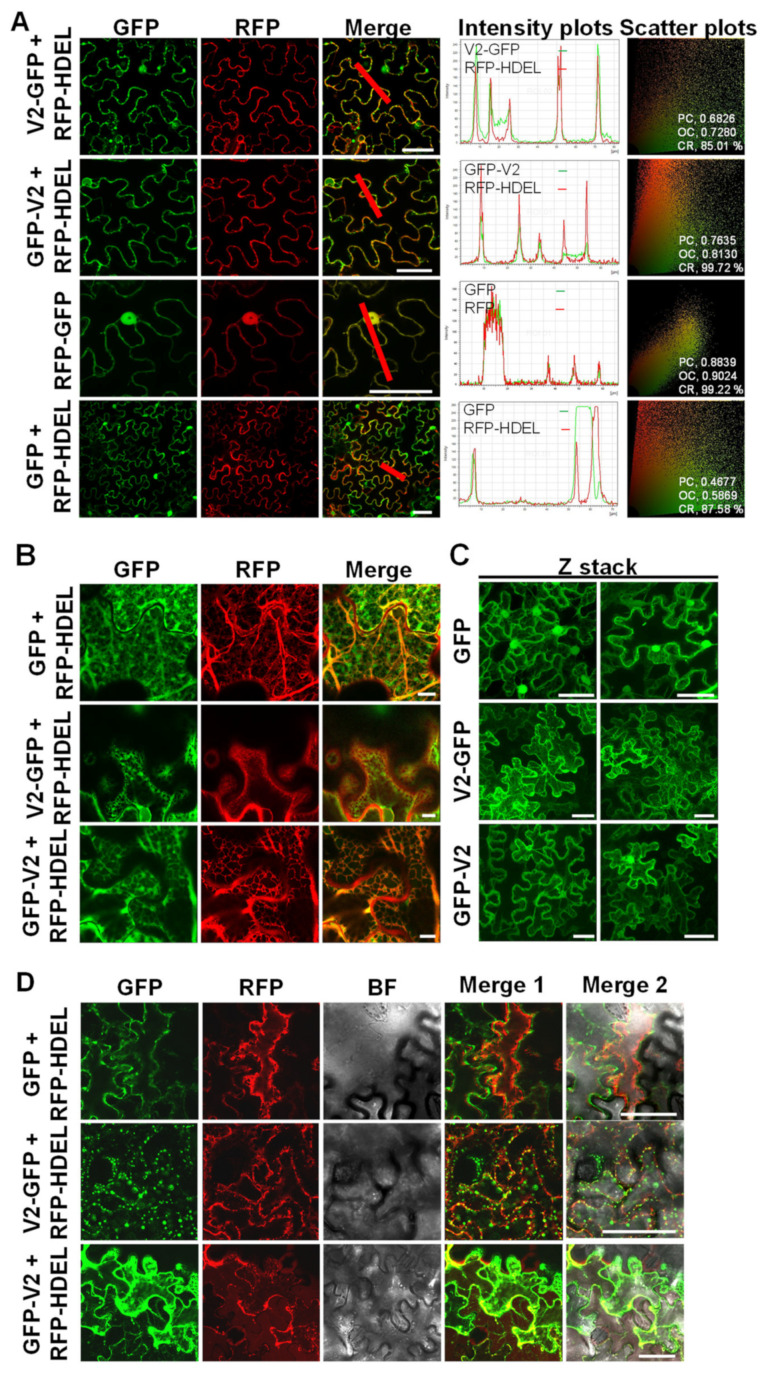
Subcellular localization of GFP-fused V2 proteins in *N. benthamiana* epidermal cells. (**A**) Colocalization analysis of GFP-fused V2 and the ER marker RFP-HDEL. *N. benthamiana* leaves were infiltrated with *A. tumefaciens* carrying constructs to express a fused protein RFP-GFP, as positive control, or co-infiltrated with *A. tumefaciens* carrying constructs expressing RFP-HDEL and V2-GFP, GFP-V2, or free GFP as a negative control. Protein accumulation was observed under the confocal microscope at two days post infiltration (dpi). Scale bar: 50 μm. Colocalization between the GFP and RFP channels is analyzed with intensity plots (under the red line of the merge image) and scatter plots. (PC: Pearson’s Correlation, OC: Overlap Coefficient, CR: Co-localization Rate.) (**B**) Close-up of co-expressed free GFP, V2-GFP, or GFP-V2, with RFP-HDEL. Bar: 50 μm. (**C**) Maximum projection of Z-stack images of free GFP, V2-GFP, and GFP-V2. Scale bar: 50 μm. Videos for the Z-stack images can be found as Supplementary Movies S1–S6. (**D**) Plasmolysis in *N. benthamiana* epidermal cells. *N. benthamiana* leaves were co-infiltrated with *A. tumefaciens* carrying constructs expressing RFP-HDEL and GFP, V2-GFP, or GFP-V2. Plasmolysis of *N. benthamiana* leaves was observed under the confocal microscope 15 min after treatment with 1 M NaCl. Scale bar: 50 μm. BF, bright field. Hechtian strands are observed in GFP-V2 samples during plasmolysis, but not in free GFP samples. These experiments were repeated three times with similar results.

**Figure 2 viruses-14-02804-f002:**
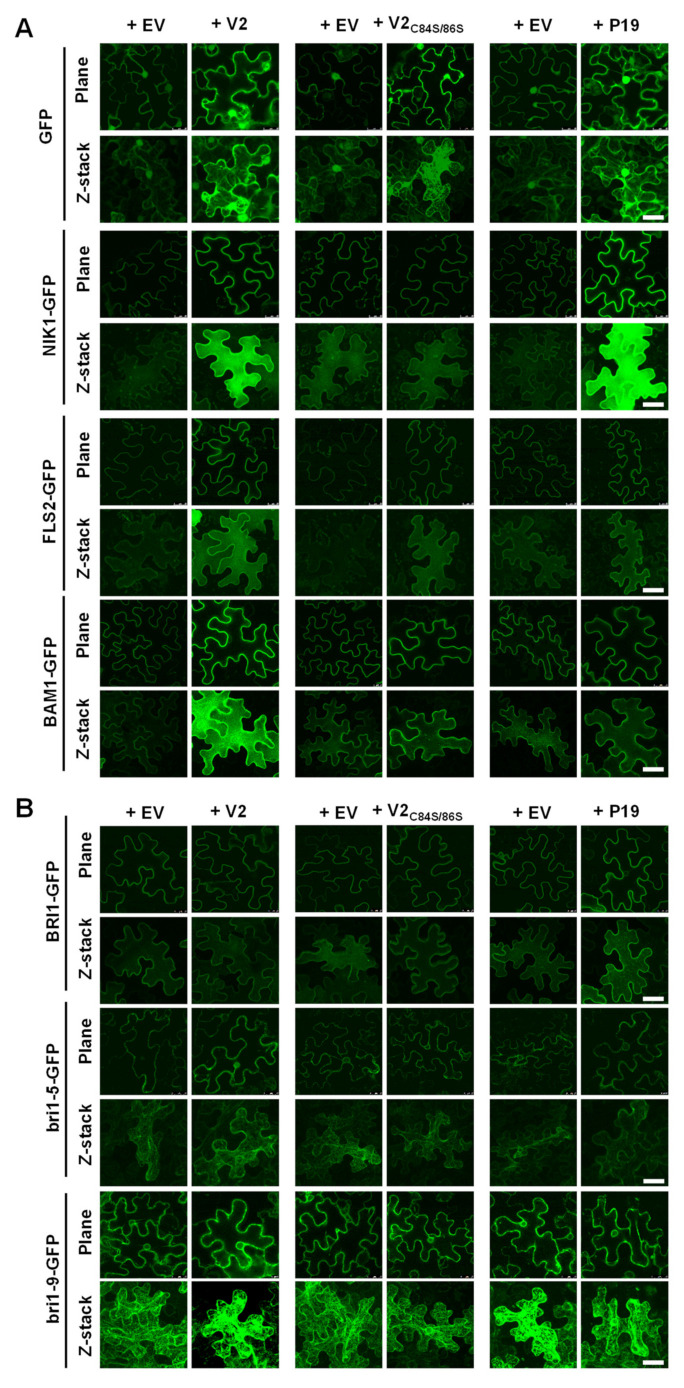
V2 does not interfere with the subcellular localization of plasma membrane-localized plant receptor-like kinases. *N. benthamiana* leaves were infiltrated with *A. tumefaciens* carrying constructs to express V2, V2_C84S/C86S_, or p19, and GFP, NIK1-GFP, FLS2-GFP, or BAM1-GFP (**A**) and BRI1-GFP, bri1-5-GFP, or bri1-9-GFP (**B**), respectively; *A. tumefaciens* carrying an empty vector (EV) were used as negative control. Protein accumulation was observed under the confocal microscope at 2 days post-infiltration (dpi). The RNA silencing suppressor from tomato bushy stunt virus (TBSV) p19 (Qu and Morris, 2002) was used as a control. EV: empty vector. This experiment was repeated three times with similar results; representative images are shown. Scale bar: 50 μm.

**Figure 3 viruses-14-02804-f003:**
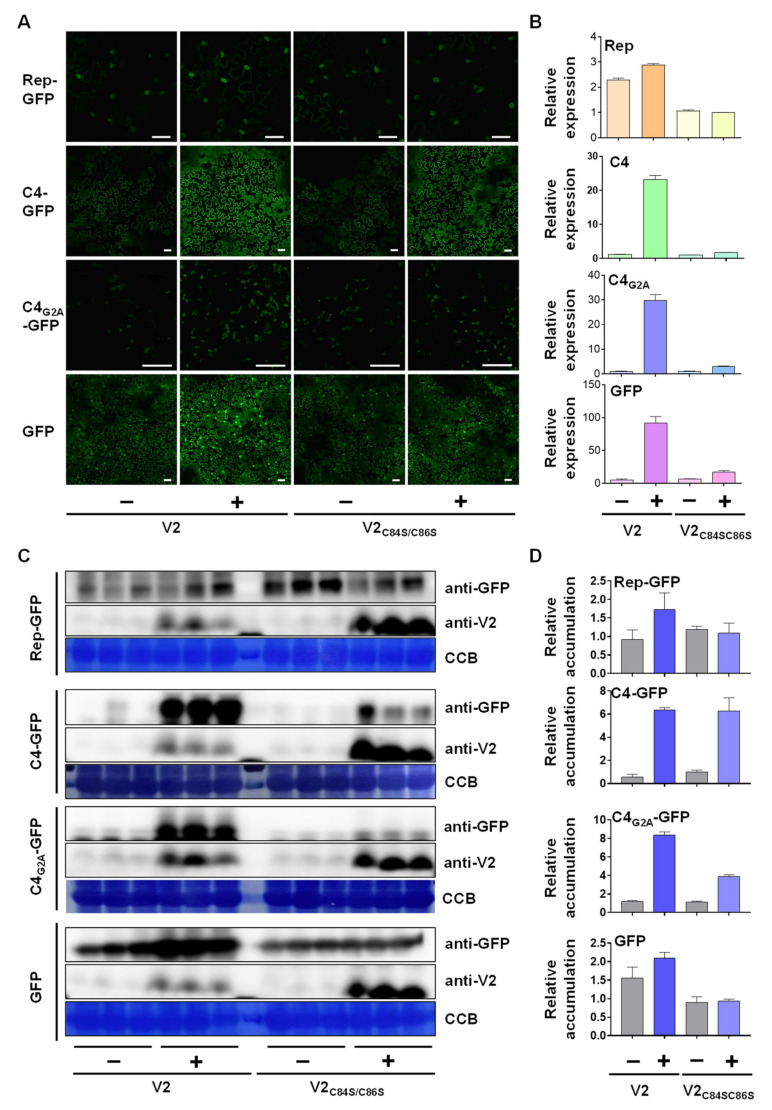
V2 increases the accumulation of the plasma membrane-localized viral protein C4 independently of its silencing suppression activity. *N. benthamiana* leaves were co-infiltrated with *A. tumefaciens* carrying constructs to express Rep-GFP, C4-GFP, C4_G2A_-GFP, or GFP, and V2 or V2_C84S/C86S_, or carrying an empty vector (EV) as negative control. The fluorescence signal was detected under the confocal microscope (**A**), and mRNA accumulation was determined by RT-qPCR at 2 days post-infiltration (dpi) (**B**). Gene expression was normalized to *NbEF1a*. Values are the mean of three independent biological replicates; error bars indicate SEM. These experiments were repeated three times with similar results; in (**A**), representative images are shown. Scale bar: 50 μm. Protein accumulation was detected by Western blot (**C**), and the relative abundance of Rep-GFP, C4-GFP, C4_G2A_-GFP, or GFP was analyzed by Image J (**D**). Values are the mean of three biological replicates; error bars indicate SEM.

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
