# Peer review of "The V2 Protein from the Geminivirus Tomato Yellow Leaf Curl Virus Largely Associates to the Endoplasmic Reticulum and Promotes the Accumulation of the Viral C4 Protein in a Silencing Suppression-Independent Manner"

_viruses, 2022, doi:10.3390/v14122804_

Round 1

Reviewer 1 Report

Wang and colleagues investigated in their study the strongest silencing suppressor encoded by the geminivirus tomato yellow leaf curl virus (TYLCV), namely V2. V2 suppresses both PTGS and TGS, is located in punctate cytoplasmic structures and in the nucleus, but also at the ER. The results pinpoint to the fact that V2 has no obvious influence on the localization of ER-synthesized receptor-like kinases (RLKs) or the ER quality control (ERQC)/ER-associated degradation (ERAD). Wang and colleagues furthermore used the V2C84S,86S mutant, which abolishes its silencing suppression activity and by this revealed a novel function of V2:  to enhance accumulation of the viral C4 protein, but independent of the capacity to suppress RNA silencing.
The techniques are described in detail, figures are intuitive and of good quality. English is good, the manuscripts reads well.
In summary, the manuscript fulfills all requirements for publication, because the manuscript presents new and interesting data, but the reviewer would like to pinpoint to the following:
-    Fig. 3B , protein expression of V2 (C84 and 86S) is much lower than V2 (wt) in the co-inculation experiments with C4 (G2A9-GFP and GFP. Especially for the GFP control the reviewer would like to see equal expression of V2 (wt) and V2 (C84 and 86S) in the Western blot. The reviewer recognizes it is equal in C4-GFP and Rep-GFP, which supports the conclusion, but the controls must be repeated / it must be shown that V2 has not a general effect on protein expression levels, but specifically for C4.
-    In addition I would like to suggest to test V2 in its ability to suppress ER stress by ion leakage experiments with tunicamycin, which could explain the observation, but only if the authors have the chance to do it.
The reference list covers the relevant literature adequately and in an unbiased manner.
Minor comments:
l. 167, delete “Bulleted”
l.168 maybe controversial than contested: …and may be involved in viral movement [23], although this is controversial [24].
l. 174 maybe  our usual model plant instead of study system; if system then patho-system
please describe in 3.1. why you performed / show plasmolysis
Fig 1: make it bigger, that it fits the page. Please provide the z-stacks as movies in supplm., if files are not too big.
